# Lightweight and Flexible Graphene Foam Composite with Improved Damping Properties

**DOI:** 10.3390/nano12081260

**Published:** 2022-04-08

**Authors:** Tong Li, Juan Du, Mi Xu, Zhuoyu Song, Mingfa Ren

**Affiliations:** 1Department of Engineering Mechanics, Dalian University of Technology, Dalian 116024, China; tong@dlut.edu.cn (T.L.); dujuanaidagong@mail.dlut.edu.cn (J.D.); rice_dlut@mail.dlut.edu.cn (M.X.); song_zhuoyu@mail.dlut.edu.cn (Z.S.); 2State Key Laboratory of Structural Analysis for Industrial Equipment, Dalian University of Technology, Dalian 116024, China

**Keywords:** graphene oxide foam, PDMS, loss modulus, damping

## Abstract

As an elastomer, PDMS can effectively suppress vibration in various fields in a certain temperature range by its viscoelastic behavior in the vitrification transition region, but the vibration isolation effect is poor at high temperature. In this paper, a three-dimensional graphene oxide (GO) foam is fabricated by solution processing method and freeze-drying techniques. After sequential infiltration synthesis, a GO-foam-reinforced PDMS nanocomposite (GO/PDMS) is fabricated with improved damping ability. By adjusting the content of GO, the micros-tructure of GO foam can be sensitively changed, which is crucial to the damping properties of composites. In this paper, by the dynamic mechanical analysis (DMA) of pure PDMS and five kinds of GO/PDMS composites, it is proved that the GO/PDMS composites developed in this work have reliable elasticity and viscoelasticity at 25 °C, which is 100 °C higher than the applicable temperature of pure PDMS. The storage modulus can reach 3.58 MPa, and the loss modulus can reach 0.45 MPa, which are 1.87 times and 2.0 times of pure PDMS, respectively. This GO-based nanocomposite is an ideal candidate for damping materials in passive vibration isolation devices.

## 1. Introduction

Shock absorption is a key challenge in many important applications, especially for the protective mechanical support of devices and instruments in aircraft, where vibration will not only seriously affect the accuracy of these electrical instruments but also shorten the service life of these devices [1,2].

To solve the above problems, passive vibration isolation systems are invented to achieve shock absorption purposes. A passive vibration isolation system provides a designer with a range of stiffness and damping characteristics, which has become one of the most important ways to reduce the vibration of instruments because of advantages such as simple structure, easy implementation, reliable operation, and no additional external energy consumption [3,4]. The performance of a passive isolation system is closely related to the damping property of materials [5,6]. Therefore, it is critical to improve the damping performance of isolation materials that have both outstanding mechanical carrying ability and improved damping performance. 

Due to its highly specific surface area and lightweight characteristics, graphene oxide (GO) is similar to graphene for engineering applications. However, in the inter-layer structure of GO, there are a large number of oxygen-containing functional groups that modify the carbon atoms, enlarging the interlayer distance and the material and making it is easier to disperse in solvents [7]. Moreover, GO is also easy to disperse in other polymer matrices, significantly improving the elastic modulus, tensile strength, electrical conductivity, and thermal stability compared with the original polymer [8,9,10,11,12]. The size of the GO flakes can be appropriately adjusted according to the application, ranging from nanometers to millimeters [13]. Due to these specific characteristics of GO, including its highly specific surface area, adjustable flake size, and light weight, and the low price of its raw materials, GO is widely investigated for applications, including electronic devices (biosensors and transparent electrodes, etc.), biomedicines [14,15,16,17,18], etc.

For the use of GO as structural materials, many efforts have been made to develop GO-based composites with ideal mechanical properties in the last decade [19,20,21]. The authors have investigated the physical mechanisms of energy absorption, while graphene and GO are used to reinforce engineering plastics [22]. However, the use of GO in shock-absorbing materials is still limited due to the low fracture toughness as a result of the addition of GO. Polydimethylsiloxane (PDMS) is a high molecular hydrophobic polymer with much higher fracture toughness compared to engineering plastics, in which the silicon–oxygen bond serves as main chains and organic functional group serves as side chains. The linear molecular structure and organic functional group side chains endow PDMS with unique flexibility and viscoelasticity, which enables the PDMS to convert the energy from the external loading into internal energy after overcoming inter-molecular friction and to finally get dissipated [23]. Therefore, PDMS has outstanding damping properties and has been investigated for energy-absorption applications. The viscoelastic properties of PDMS are beneficial to the passive vibration isolation in mechanical systems, making PDMS an ideal candidate material for shock-absorption materials [24,25,26]. Zhang et al. prepared isotropic magnetorheological elastomers with GO-filled polydimethylsiloxane as a matrix via the solution-blending casting method. The results showed that the addition of GO sheet hindered the chemical cross-linking of PDMS matrix, resulting in gaps between GO sheet and PDMS matrix. The loss modulus of the composite was improved due to the interface weakening between GO and PDMS, and the aggregation of GO [27]; Fang et al. proposed a simple and universal method to bond graphene foam (GF) with PDMS to prepare a corresponding composite material. The results show that the composite material has significantly improved mechanical properties and thermal stability compared with pure PDMS and GF/PDMS [8]; Han et al. fabricated boron nitride nanosheets (BNNS)/epoxy composites by constructing a simulated 3D conductive network in the epoxy resin matrix using a bi-directional freezing technique. When boron nitride content is low (15vol%), the composite has excellent resistivity and thermal stability [28]; Peter et al. injected a certain amount of PDMS into the freeze-drying foam of GO, which improved the mechanical properties of the composite material. Furthermore, the interaction mechanism between the GO layer and PDMS was further elaborated through molecular dynamics simulation, which proved the stability of the composite material at high temperature [29]; Min et al. used a bi-directional freezing technology to control the growth direction and rate of ice crystals in the early freezing stage of GO suspension and obtained three different GO foam structures [30].

In this work, a three-dimensional GO foam is prepared by solution methods, followed by a sequential infiltration synthesis, and a 3D GO/PDMS nanocomposite is fabricated. The service temperature of GO/PDMS nanocomposite with high loss modulus can be increased from 150 °C to 250 °C, and both the storage and loss modulus of the nanocomposite change significantly compared to pure PDMS.

## 2. Materials and Methods

### 2.1. Preparation of Graphene Oxide

Graphene oxide (GO) was prepared by a modified Hummer method. A 9:1 mixture of concentrated H_2_SO_4_/H_3_PO_4_ (360 mL: 40 mL) was added to graphite powder (3.0 g), KMnO_4_ (18.0 g), which was slowly added to the above-mixed solution. The solution was heated to 50 °C and stirred for 12 h. After the reaction cooled down to room temperature, H_2_O_2_ was added to the mixed solution until the solution turned golden yellow, and the mixture was filtered. The remaining solid matter was washed with deionized water, HCL, and ethanol in sequence [31]. The materials were later washed by deionized water (to be weak acidity) and centrifuged at a speed of 4000 rpm for 45 min. The remaining solid matter was dissolved in deionized water, subjected to ultrasonic operation to obtain a suspension solution, and then freeze-dried to obtain ~5.8 g GO.

### 2.2. Preparation of GO Foam

100 mg of GO powder was stirred and sonicated in 10 mL of water, 2 µL of glutaraldehyde, and 20 mg of resorcinol, to make the solution evenly mixed. The solution was then poured into a plastic petri dish, to be frozen in liquid nitrogen and dried in a freeze dryer to obtain the 3D GO foam [29]. Five different GO foams were fabricated with different concentrations of the GO solution; these foams were C100, C150, C200, C250, and C350, representing a GO solution concentration of 100mg/ 30mL, 150 mg/30 mL, 200 ng/30 mL, 250 mg/30 mL, and 350 mg/30 mL, respectively.

### 2.3. Preparation of GO/PDMS Composite

PDMS and curing agent are mixed uniformly with a weight ratio of 10:1, to obtain a PDMS precursor. The alkane precursor was dropped into the graphene oxide foam, infiltrated at room temperature in a vacuum oven for 2 h at room temperature, and cured at 85 °C for 3 h to obtain the GO composite material [30]. The nanocomposites with different GO foams are marked as C100, C150, C200, C250, and C300, respectively, which reflects the GO foam employed. Figure 1 illustrates the 3D composite fabrication process.

### 2.4. GO Characterization

Raman spectroscopy is widely used to detect the degree of structural disorder of graphene. In this work, a Confocal Raman microscope (inVia Qontor, UK) is employed to validate the successful preparation of GO materials. A 633 nm He-Ne laser was used to scan the GO powder prepared in this study, and the scanning range was 1000 cm^−1^~4000 cm^−1^.

### 2.5. SEM Characterization

The surface morphology of the GO and GO/PDMS nanocomposites was characterized by field emission scanning electron microscopy (NOVA Nano SEM 450, FEI Company, Hillsboro, OR, USA). Different samples were prepared with different GO contents in the solution-processing method, and three samples for each group of composite materials were prepared. The size of each sample was 3 mm × 3 mm × 3 mm, and the sample surfaces needed to be flat and clean. 

### 2.6. DMA Characterization

The loss modulus, storage modulus, and loss factor of the composites with different graphene oxide contents were characterized using a dynamic mechanical analyzer (DMA Q800, TA Instruments, New Castle, DE, USA); 1% loading strain was applied to the materials; the temperature range was 25–300 °C; the heating rate was 5 °C/min; and the diameter of the circular sample was 8 mm.

## 3. Results and Discussion

Five different GO foam structures were fabricated by adjusting the concentration of GO solution in the synthesis process. The morphology and viscoelastic properties were characterized.

### 3.1. Morphology of GO/PDMS

Figure 2 shows the Raman spectrum of the GO prepared in this work, in which the Raman spectrum of single-layer GO is in the range of 1000–1800 cm^−1^. It can be seen that the prepared material exhibits two peaks, which are the D peak at 1350.50 cm^−1^ and the G peak at 1591.64 cm^−1^. G peak is the characteristic of sp^2^ hybridized carbon and has an asymmetric shape; the existence of D peak is one of the most important features of GO, and it showed that the sp^2^ carbon lattice was strongly destroyed and disordered during the preparation of GO. The intensity ratio (I_D_/I_G_) of peak D to peak G is used to characterize the disorder degree of GO. The larger I_D_/I_G_ is, the more disordered structures are introduced in the oxidation process.

Figure 3a–c is the SEM image of the graphene oxide foam in three different directions. It can be seen that in the XY plane, the GO foam has an irregular porous structure. In the XZ and YZ planes, the GO foam has a regular layered structure, which is beneficial to the impact resistance and the capacity of the energy dissipation of the foam material; as the content of GO increases, the foam structure becomes more compact. Figure 3d–f is the SEM image of the GO/PDMS composite in three directions. It can be seen that during the PDMS filling process, the original GO foam structure remains intact, and the regular structure in the z-direction formed by dry-freezing is conducive thoughout the filling process of PDMS. After the infiltration of the PDMS, the GO foam structure and the PDMS interface are well connected, and limited amounts of defects can be observed.

### 3.2. Dynamic Mechanical Properties of GO/PDMS

Dynamic mechanical analysis (DMA) was performed on the GO/PDMS composites perpendicular to the laminar direction and compared with pure PDMS in this section. 

Taking the C200 composite an as example, in Figure 4a, the storage modulus and loss modulus of GO/PDMS, with different GO contents in the temperature range of 100–250 °C, are provided. The results show that without the addition of GO foam, the loss factor of PDMS continues to decrease with temperature in the whole characterized temperature range, which means the viscoelastic properties of pure PDMS are limited at high service temperatures. After the addition of GO foams, the storage modulus and loss modulus remain saturated below 150 °C and gradually increase above 150 °C, until 250 °C, which indicates a reliable viscoelastic property in the temperature range of 150–250 °C.

Figure 4b shows the loss factor of nanocomposites with different GO foams. It can be found that, for the samples with a GO concentration of 100 mg/30 mL, the loss modulus keeps decreasing in the whole temperature range. For the other four types of GO/PDMS composites, the loss factor all increased after around 150 °C and became higher than pure PDMS. The sample prepared by 250 mg/30 mL GO solution shows a most outstanding loss factor, which is 49.0% higher than pure PDMS material.

The storage modulus reflects the energy stored in the composites, and the storage modules of these composites are provided in Figure 4c. It can be found that, except for the C100 composite, all the other samples show higher storage modulus than pure PDMS in the temperature range after 100 °C, which means that the GO/PDMS nanocomposites have a higher mechanical bearing ability compared to pure PDMS.

The loss modulus represents the viscous part of the amount of energy dissipated in the composite. Figure 4d provides the information of loss modulus in the different nanocomposites prepared in this work. It can be found that, when the GO content during solution synthesis is higher than 100 mg/30 mL, the loss modulus of the nanocomposites is significantly higher than pure PDMS, whose physical mechanisms have been explained in our previous research [30,32]. For the C250 sample, the loss modulus reaches a peak value of 0.45 MPa, which increases the loss modulus of pure PDMS by two times.

For the C100 composite, it is difficult to form a complete and stable layered foam skeleton as the concentration of GO is low in the solution synthesis process, which causes a reduction of both storage and loss modulus. With the increase of GO content in the foam skeleton, the loss factor, storage modulus, and loss modulus of the composite material all increase [27]. When the content of GO increases to 350 mg/30 mL during the solution synthesis process, the GO content is too large, the GO foam structure reaches a saturated state in the solvent, and the obtained GO foam skeleton is too stiff and reduced dynamic viscoelasticity of the material. In Figure 4e, the differences in storage modulus, loss modulus, and loss factor of different GO/PDMS composites can be seen intuitively, which shows that the overall performance of C250 is excellent. Storage modulus, loss modulus, and loss factors are all summarized in Table 1, and it can be found that when the graphene oxide content reaches 250 mg/30 mL during the synthesis process, the storage modulus of the GO/PDMS composite prepared can reach 3.58 MPa, the loss modulus can reach 0.45 MPa, and the loss factor can reach up to 0.125, which constitutes the most outstanding overall performance with both mechanical bearing ability and viscoelastic performance.

In the past studies, many methods have been applied to the preparation of GO-based composites, such as the bidirectional freezing method to construct 3D GO foam structures, and the construction of 3D foam skeletons that are fiber-reinforced [28,32]. By effectively controlling the density of the 3D porous foam structure, the GO/PDMS composites prepared in this study can sensitively improve the damping properties and be used in a wider temperature range. The service temperature of our newly fabricated 3D GO/PDMS nanocomposite can be increased to 250 °C, which is almost 100 °C higher than pure PDMS base material.

## 4. Conclusions

In the present study, five highly aligned porous graphene oxide (GO) foams are synthesis by solution method and freeze-drying techniques. After infiltration of polydimethylsiloxane (PDMS), a 3D GO/PDMS nanocomposite can be fabricated with improved viscoelastic properties while still keeping the mechanical stiffness. The influences of GO content are discussed in terms of the microstructure and dynamic mechanical properties. The results show that there is no significant differences in the structures of GO foams with different contents. This is because unidirectional freezing technology is adopted in solution freezing, and the interface of GO foam skeleton and PDMS is well combined, which is conducive to improving the mechanical properties of the composites. In the high temperature range of 150–250 °C, the storage modulus, loss modulus, and loss factor of GO/PDMS composites increase with the increase of GO content. However, when the GO content is higher than 250 mg/30 mL, both the loss modulus and loss factor decrease. This is due to the fact that with the increase of go content, the filler is more likely to aggregate and the stiffness of the composite is improved, which leads to an increase in storage modulus and a decrease in loss modulus. Compared with pure PDMS, the storage modulus and loss modulus of the composites filled with GO increased. Especially, the C250 composite achieved the largest increase: its storage modulus and loss modulus were 3.58 MPa and 0.45 MPa, which were 1.87 and 2.0 times, respectively. This lightweight 3D nanocomposite has the advantages of ideal mechanical bearing ability and high efficiency of energy absorption. These characteristics make this 3D nanocomposite an ideal candidate for a damping element in passive isolation systems in the design of high-performance equipment that faces severe vibration conditions.

## Figures and Tables

**Figure 1 nanomaterials-12-01260-f001:**
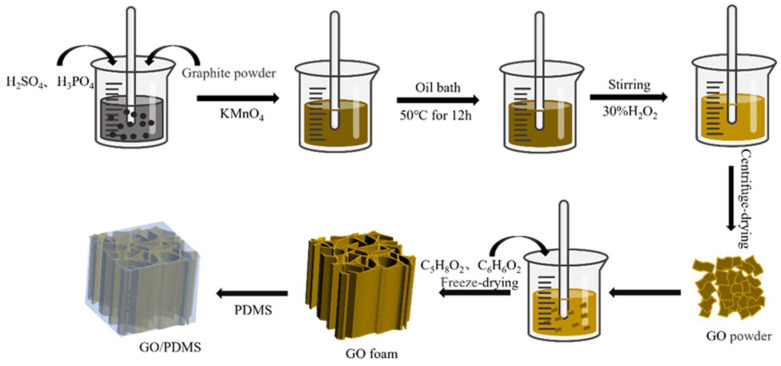
The fabrication process of 3D GO/PDMS composite.

**Figure 2 nanomaterials-12-01260-f002:**
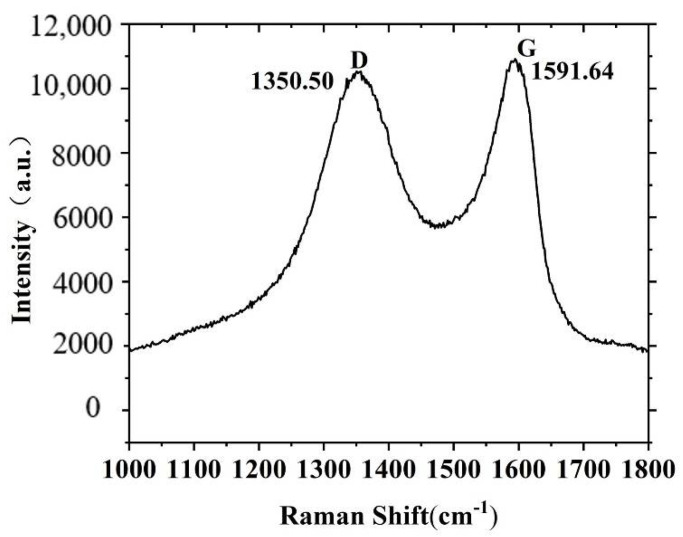
Raman spectrum of GO prepared in this study.

**Figure 3 nanomaterials-12-01260-f003:**
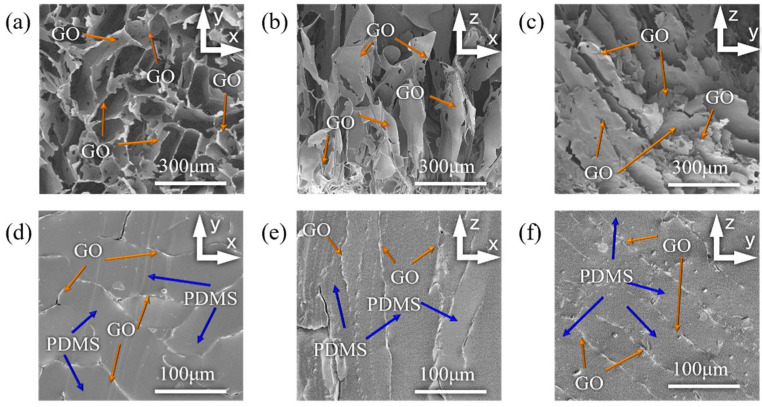
Morphology of GO foam and 3D GO-PDMS composite. (**a**–**c**) shows the three-dimensional structure of GO foam in three directions; (**d**–**f**) shows that the PDMS matrix is well infiltrated into the skeleton of GO foam in all directions.

**Figure 4 nanomaterials-12-01260-f004:**
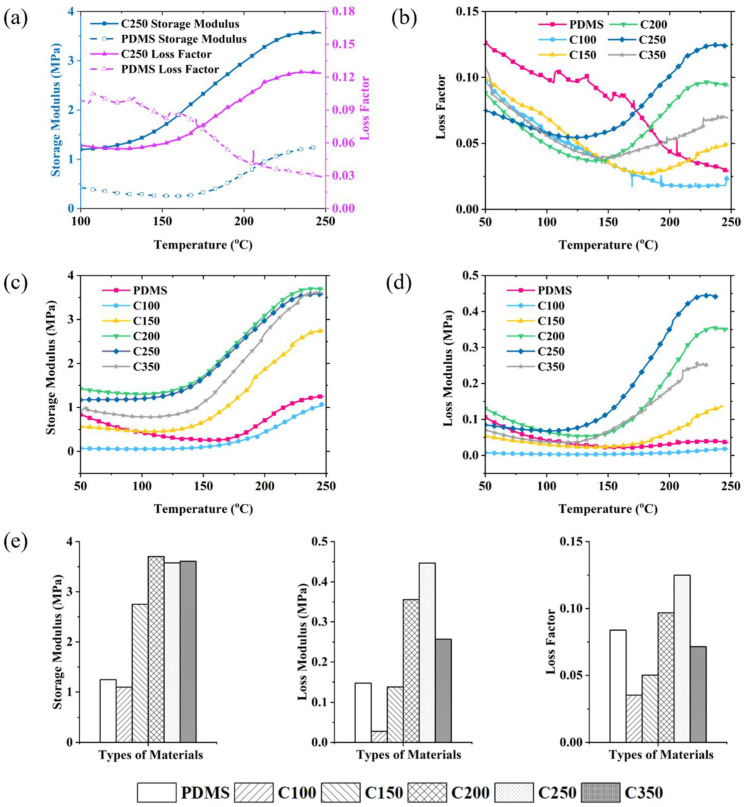
Viscoelastic properties of different GO/PDMS composites. (**a**) is the comparison of storage modulus and loss factor between C250 composite and pure PDMS. (**b**–**d**) provide information on the loss factor, storage modulus, and loss modulus of different GO/PDMS composites. (**e**) summarizes the comparison of the storage modulus, loss modulus, and loss factor of different GO/PDMS composites, in the temperature range of 150–250 °C.

**Table 1 nanomaterials-12-01260-t001:** The storage modulus, loss modulus, and loss factor of different GO/PDMS composites.

Types of Materials	PDMS	C100	C150	C200	C250	C350
Storage Modulus (MPa)	1.24775	1.09806	2.75027	3.70336	3.57718	3.60997
Loss Modulus (MPa)	0.14733	0.0277	0.13773	0.35549	0.44676	0.25675
Loss Factor	0.08384	0.03521	0.05014	0.09686	0.1249	0.0715

The above data are the peak values of different GO/PDMS composites in the temperature range of 150–250 °C.

## Data Availability

The data presented in this study are available in this paper.

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
