# Peer review of "Lightweight and Flexible Graphene Foam Composite with Improved Damping Properties"

_nanomaterials, 2022, doi:10.3390/nano12081260_

Round 1

Reviewer 1 Report

The manuscript "Lightweight and flexible Graphene Foam Composite with Improved Damping Properties" reports a sound piece of research on the use of s three-dimensional graphene oxide (GO) foam reinforced by PDMS nanocomposite (GO/PDMS) as a tool to improve the ability of damping. The results reported are of potential interest for readers of nanomaterials journal and worthy of publication, but before several important issues should be clarified: the ABSTRACT is not clear, It doe not explain exactly the focus of the manuscript; the INTRODUCTION does not adequately explain why the authors chose to improve and potential the GO; the MATERIALS and METHODS are well written; the CONCLUSION needs to be rewritten because does not justify the results obtained and does not explain the possible use of this new nanomaterial; the REFERENCES are adequate. Furthermore, I also recommend a minor revision of the manuscript. 

Author Response

Dear reviewer, I have rewritten the ABSTRACT and CONCLUSION  according to your requirements, and made some supplements in the INTRODUCTION. The revised contents have been marked yellow in the article, please see the attachment.

Reviewer 2 Report

The manuscript entitled “Lightweight and flexible Graphene Foam Composite with Im- 2

proved Damping Properties” has been submitted by authors. Some issues to be addressed which will improve the quality of manuscript. Therefore, I recommend this work could be published after the major revision

  1. The English composition requires many improvements. The authors should proofread the manuscript carefully to minimize grammatical errors.
  2. The background of this work is not clear. The authors should specify in a clearer way what novel and original this work proposes to readers based on some new works. Author please added comparative table for reader clear understanding.
  3. In Fig. 2 X -axis unit look incomplete, please correct it.
  4. In Fig. 4 X -axis unit is missing.
  5. The characterization part and the result and discussion part are not supported by enough references. It may be supported by the recent relevant references

Journal of Colloid and Interface Science, 589, 2021, 401-410; Rheologica Acta, 61, 215–228, (2022)

Author Response

Dear reviewer, I have made some additions to the INTRODUCTION according to your suggestions. The revised content has been marked yellow in the article, checked some grammatical errors, and corrected the x-axis units in Figure 2 and Figure 4. The two references you provided are cited in the introduction, which are No. 18 and No. 27 respectively. Please see the attachment.

Reviewer 3 Report

Comment 1. Make graphical figures bold letter as its not visible properly.

Comment 2. Mention about D and G band of GO raman data, and discuss it.

Author Response

Dear reviewer, I have bolded the graphic font and enlarged the font size according to your suggestion, and discussed the D peak and G peak mentioned in the Raman spectrum of GO. The revised content has been marked yellow in the article, please see the attachment.

Round 2

Reviewer 2 Report

The author done all the suggested changes and now it's suitable for publication in nanomaterials